# Magnetic-Field-Assisted Molecular Beam Epitaxy: Engineering of Fe_3_O_4_ Ultrathin Films on MgO(111)

**DOI:** 10.3390/ma16041485

**Published:** 2023-02-10

**Authors:** Adam Dziwoki, Bohdana Blyzniuk, Kinga Freindl, Ewa Madej, Ewa Młyńczak, Dorota Wilgocka-Ślęzak, Józef Korecki, Nika Spiridis

**Affiliations:** 1Jerzy Haber Institute of Catalysis and Surface Chemistry, Polish Academy of Sciences, Niezapominajek 8, 30-239 Krakow, Poland; 2PREVAC sp. z o.o., Raciborska Str. 61, 44-362 Rogów, Poland

**Keywords:** magnetite, ultrathin epitaxial films, magneto-optic Kerr effect, conversion electron Mössbauer spectroscopy, external magnetic field, magnetic anisotropy

## Abstract

Molecular beam epitaxy is widely used for engineering low-dimensional materials. Here, we present a novel extension of the capabilities of this method by assisting epitaxial growth with the presence of an external magnetic field (MF). MF-assisted epitaxial growth was implemented under ultra-high vacuum conditions thanks to specialized sample holders for generating in-plane or out-of-plane MF and dedicated manipulator stations with heating and cooling options. The significant impact of MF on the magnetic properties was shown for ultra-thin epitaxial magnetite films grown on MgO(111). Using in situ and ex situ characterization methods, scanning tunneling microscopy, conversion electron Mössbauer spectroscopy, and the magneto-optic Kerr effect, we showed that the in-plane MF applied during the reactive deposition of 10 nm Fe_3_O_4_(111)/MgO(111) heterostructures influenced the growth morphology of the magnetite films, which affects both in-plane and out-of-plane characteristics of the magnetization process. The observed changes are explained in terms of modification of the effective magnetic anisotropy.

## 1. Introduction

Modern spintronics requires nano- and heterostructures with controllable and programmable magnetic properties [1]. Molecular beam epitaxy (MBE), a widely used method of growing magnetic heterostructures [2], enables a certain level of indirect control of the magnetic properties. The primary factor influencing the properties of epitaxial heterostructures is the growth mode, which determines the resulting atomic- and microstructure. The structural properties depend on the elementary steps of the epitaxial growth, which include adsorption/desorption, surface diffusion, nucleation/coalescence, and, finally, film growth. In traditional MBE technology, these elementary processes are controlled by the substrate temperature, deposition rate, and partial pressure of reactive gases [3]. Additionally, external agents can be used, such as plasma generation [4] or ion beams [5]. However, other factors, including the external stimuli proposed in the present study, have been only occasionally applied in the physical vapor deposition of thin films. In particular, due to the limitations of the ultra-high vacuum (UHV) technology, examples of applying external fields in situ during growth, as well during in situ post-deposition treatment, are scarce [6], and to the best of our knowledge, there are no examples of UHV MBE under an external magnetic field.

However, several studies have reported the deterministic influence of an external magnetic field on the composition, crystal structure, and magnetic properties of thin oxide and metal films prepared via different methods. Kim et al. [7] found that a continuous Ni catalyst layer on Si(001) was dewetted by post-deposition annealing and agglomerated into well-separated dots, the size and distribution of which were strongly affected by the moderate magnetic field. The microstructure of BiFeO_3_ epitaxial thin films was modified by the application of a magnetic field during pulsed laser deposition (PLD): a columnar structure was shown in the film prepared under a high deposition rate for a magnetic field of 0.4 T [8]. Nilsen et al. [9] studied the effect of the magnetic field on the atomic layer deposition growth of hematite films and observed distinct growth modifications partially ascribed to the weak ferromagnetism of α-Fe_2_O_3_, but they did not explain the physics behind it. Zhang et al. [10] developed a system for PLD in a high magnetic field, up to 10 T, and they showed that the epitaxial growth of oxide nanostructures could be tuned from continuous films to nanorods, which caused changes in the magnetic anisotropy. In the close material relation to the present paper, Stadler et al. [11] studied the role of applied magnetic fields (0.5 T) during a chemical vapor deposition (CVD) of iron oxides and observed the essential influence of the magnetic field on phase composition, morphology, and magnetic properties.

In summary, whereas post-deposition annealing under an external magnetic field at high temperatures is a routine procedure for shaping desired magnetic anisotropy (especially for multicomponent alloy and compound films [12]), the engineering of magnetic properties using magnetic-field-assisted (MF-assisted) MBE remains unexplored.

Therefore, we undertook systematic studies of MF-assisted MBE for different epitaxial systems ranging from metallic ultrathin films through oxide layers and more complex heterostructures (e.g., ferromagnet-antiferromagnet systems showing exchange bias), and in the present paper, we present the first results on MF-assisted epitaxial growth of magnetite films on MgO(111).

## 2. Materials and Methods

### 2.1. Properties of Epitaxial Magnetite Films

Magnetite (Fe_3_O_4_) is one of the oldest known magnetic materials. In recent years, magnetite has acquired increased interest from researchers not only due to being a strongly correlated electron system, but also due to being an attractive material for the nascent field of spintronics because of its half-metallic character, high conductivity, and high Curie temperature (for a review, see [13]).

Magnetite has an inverse spinel structure with Fe^3+^ ions in the tetrahedral sublattice (so-called A-sites) and the mixed-valency Fe^2+/^Fe^3+^ octahedral sublattice (so-called B-sites). Put simply, electron hopping averages the ionic charge in the octahedral sublattice to 2.5, and thanks to the hopping conductivity, magnetite has metal-like electric conductivity at room temperature (RT). At low temperatures below 122 K, a structural Verwey transition occurs (for a critical review, see [14]), accompanied by freezing of the electron hopping, and magnetite becomes a narrow gap insulator.

Magnetite is a ferrimagnet with both the Fe A- and B-sites contributing to the net magnetization that results from the antiferromagnetic coupling of both sublattices, with A- and B- sites magnetic moments antiferromagnetically aligned along the <111> axes above the Verwey temperature. The imbalance in the occupation of the A- and B-sites (8 and 16 atoms per unit cell, respectively) results in the net magnetic moment of 4.1 μ_B_ per the Fe_3_O_4_ molecule. This corresponds to the saturation magnetization M_s_ = 471 emu/cm^3^ at RT. The second-order magnetic property, i.e., the magnetocrystalline anisotropy, which refers to the energy required to rotate magnetization from a magnetically easy to hard direction, reflects the cubic symmetry of the magnetite structure: the low index (111), (100), and (110) directions are the easy, hard, and intermediate magnetization axes, respectively, and cubic the first- (K_l_) and second-order (K_2_) anisotropy constants at RT are K_1_ = −1.25(5) × 10^5^ erg/cm^3^, and K_2_ = −0.30(5) × 10^5^ erg/cm^3^ [15]. Consequently, in the multi-domain state, the magnetization is distributed between eight equivalent <111> directions of the four cubic (111)-type easy axes.

Another property important for the present study is magnetostriction, which is the strain response to magnetization changes upon the external magnetic field. The maximum (under saturation) magnetostrictive strain in magnetite along the (111) and (100) axes is highly anisotropic and amounts to λ_111_ = 72.6 × 10^−6^ and λ_100_= −19.5 × 10^−6^ [16].

Complex structural, electronic, and magnetic properties become even more intricate in epitaxial magnetite films. Epitaxial magnetite films can be grown on different substrates, both metallic, e.g., Pt [17], Ru [18], and Fe [19], or oxidic, such as Al_2_O_3_ [20,21,22,23], SrTiO_3_ [24], MgAl_2_O_4_ [12], and MgO [25,26,27,28,29,30]. Among the latter, MgO, used in the present work, has been the most exploited, and the hetero-epitaxial Fe_3_O_4_/MgO system is important, albeit not the only case of thin film modification of the bulk magnetite characteristic.

Size effects with decreasing thickness influence both structural and electronic properties, as seen in the most remarkable magnetite feature, i.e., in Verwey transition, which becomes broader, shifts to lower temperatures, and eventually, may even vanish [24,31,32].

In ultra-thin films with a thickness below a dozen nanometers, superparamagnetism is a typical feature. It was first reported using the conversion electron Mössbauer spectroscopy (CEMS) by Fujii et al. [15] for (111)-oriented Fe_3_O_4_ films grown on the (0001) sapphire surface. A similar observation was later reported for (001)-oriented films on MgO(001) [33] and interpreted as coming from the frustration of the magnetic interactions at the antiphase domain boundaries [34] (APBs) formed during the nucleation of magnetite on the MgO substrate, with a higher surface symmetry and smaller unit cell.

APBs have been frequently considered responsible for high-saturation fields observed for epitaxial magnetite films [29]. Whereas bulk single crystals saturate along the hard axis at an effective magnetic field below 1 kOe [10], some authors report that for films on MgO(001), magnetization remains unsaturated in fields as large as 70 kOe [29] or 20 kOe [35]. Difficult saturation was also reported by other authors for Fe_3_O_4_(111) on α-Al_2_O_3_(0001) [17]. This remains in contrast with moderate or even exceptionally low (down to 0.2 kOe) saturation fields for magnetite films obtained by pulsed layer deposition on (001)-oriented MgO, MgAl_2_O_4_, and SrTiO_3_ substrates [36]. Apparently, the variation of these magnetic properties results from differences in the film morphology and structure caused by different preparation methods and protocols.

The exact determination of the magnetization is non-trivial under difficult saturation, superparamagnetism, and the uncertainty of the mass determination in the case of thin films. Therefore, reports both on the strong reduction [37] and enhancement [38] in the magnetic moments should be taken with some reservation, and a thorough analysis of the experimental data shows a decrease in the magnetic moment of magnetite thin films with decreasing thickness [39]. Considerable variation is also observed in coercivity, which ranges, e.g., from app. 50 Oe to 500 Oe in 100 nm and 160 nm films grown by (PLD) on MgO(001) [40] and SrTiO_3_(100) [41], respectively.

Details of the observed anomalies depended on the particular symmetry of the magnetic anisotropy resulting from the growth direction of the epitaxial films. The dominating anisotropy terms in an ideal single-crystal magnetite film should be the shape and magnetocrystalline anisotropies. Because the shape anisotropy *K_sh_* = 2$\pi M_{s}^{2}$ = 1.39 × 10^6^ erg/cm^3^ is an order of magnitude larger than the magnetocrystalline anisotropy, the magnetization is expected to lie in the film plane, regardless of the film orientation. However, usually, this is not the case. Historically, Fujii et al. [15] characterized (111)-oriented Fe_3_O_4_ films on sapphire and concluded that out-of-plane magnetized domains were evenly distributed in all easy directions. Such an anomalous (from the thin film’s point of view) out-of-plane distribution of the magnetic moments was later found for Fe_3_O_4_ films of different orientations, (001) or (111), grown on different substrates using various deposition techniques [16,31,42,43,44]. This anomaly can be only partially explained by competing crystalline, magnetoelastic, and shape anisotropy [16], and their understanding invokes antiferromagnetic exchange coupling across APBs [29,45].

However, the out-of-plane magnetization component is not the general feature of epitaxial magnetite films, and in-plane magnetization was found for the (111)-oriented magnetite films on metallic substrates, Pt(111) [46,47] and Ru(0001) [48]. This is, to some extent, astonishing because in the Fe_3_O_4_(111) films, one of the easy axes is perpendicular to the surface, whereas the other three form an angle 70.5° from the surface normal, as compared to the (001) orientation, with no perpendicular easy axes.

The subject of the present study is epitaxial magnetite films on MgO(111). Usually, to stabilize the (111) orientation, Al_2_O_3_(0001) substrates are used with a considerable 8%-tensile misfit with magnetite, whereas the oxygen lattice on MgO(111) perfectly matches the magnetite oxygen lattice. Nonetheless, very few papers have reported growth [49,50] and the magnetic properties [51] of the Fe_3_O_4_(111)/MgO(111) system. Chichvarina et al. [45], for a 40 nm film, reported the preferred perpendicular easy magnetization axis, as documented by a relatively high remanence of 45% and coercivity of approximately 1 kOe for the out-of-plane hysteresis loop. They also noted a pronounced dependence of these parameters on the film thickness—a thickness reduction to 20 nm and less made the films magnetically softer in-plane than out-of-plane. These properties show that the Fe_3_O_4_(111) on Mg(111) are prone to different growth factors, which makes them a good candidate for the modulation of magnetic properties during MF-assisted epitaxial growth.

### 2.2. Experimental Details

The magnetite films were grown in a multi-chamber UHV system (PREVAC, Rogów, Poland) base pressure 2·10^−10^ mbar). A preparation chamber includes a home-built MBE system and typical surface characterization tools: a 4-grid optics (OCI Vacuum Microengineering Inc. London, ON, Canada) for low-energy electron diffraction (LEED) and Auger electron spectroscopy (AES). Two separate chambers are dedicated to scanning tunneling microscopy (STM, Burleigh Instruments, Fishers, NY, USA) and a home-built CEMS spectrometer. A special feature of the UHV system is the dual sample holder configuration that allows considerable versatility of the sample environment during film growth and characterization and, in particular, the possibility of applying an external magnetic field at different preparation stages. A dual sample station of a 4-axes manipulator accepts so-called “PTS” (PREVAC) and flag-style (FS) sample holders [52]. The PTS sample holders are specialized for different functions and can be operated with substrates fixed directly to the holders or as adapters accepting the FS holders carrying substrates. The range of the PTS adapters includes, among others, the holders transmitting the magnetic field from samarium–cobalt permanent magnets to a sample mounted on the FS holder, both in-plane (up to 100 mT), used in the present study, and out-of-plane (up to 200 mT). The magnetic field acting on the sample from the permanent magnets was calibrated ex situ, outside the UHV system, using a dummy substrate, with an accuracy of ±5%.

During MF-assisted deposition, the sample temperature can reach as high as 500 °C using resistive heating without degrading the Sm-Co magnets (Magnet Expert Ltd., Tuxford, UK) thanks to a water or liquid nitrogen magnet cooling system. At the preparation stages requiring even higher temperatures, the FS holders can be heated by electron bombardment in the FS holder station or a specialized PTS holder.

A series of magnetite Fe_3_O_4_(111) films were reactively grown on MgO(111) substrates (MaTeck, Jülich, Germany). Prior to deposition, the substrate was annealed at 500 °C for an hour. A home-built MBE system including Fe-isotopes and several other metal sources (e.g., Au, Co, Ni) ensured deposition of the pure elements under a 10^−10^ mbar pressure range using resistively heated BeO crucibles. For growing magnetite, the iron isotope ^57^Fe was evaporated under O_2_ partial pressure 5·10^−6^ mbar on the substrate kept at 250 °C. The magnetite film thickness of 10 nm was controlled using a quartz crystal monitor within ±5% accuracy. Each preparation run included the film deposited in the presence of the in-plane magnetic field and a reference sample with no field applied. Both films could be deposited on two halves of the same substrate. For this purpose, the substrate was transferred between the respective manipulator stations and the used half was selected by a movable shutter between the MBE evaporators and the substrate. For ex situ measurements, the magnetite films could be protected by a nonmagnetic layer, typically MgO, which was deposited using an electron beam evaporator (PREVAC).

In addition to standard UHV methods, magnetite layers were characterized using CEMS (in situ and ex situ) and the magnetooptic Kerr effect (MOKE). The geometry and detection principle of the in situ CEMS were described in Ref. [42]. For ex situ CEMS a standard Mössbauer spectrometer (Elektronika Jadrowa, Kraków, Poland [53]) and a home-built gas flow (He+10%CH_4_) proportional detector were used. The experimental CEMS spectra were numerically fitted using Voigt lines and a least square minimization procedure using the Recoil software [54].

The MOKE measurements were performed ex situ at RT using the polarization-modulation technique with a photoelastic modulator (Hinds Instruments, Hillsboro, OR, USA). For L-MOKE, the light from a 635 nm diode laser (Coherent, Santa Clara, CA, USA) illuminated the sample at 30°, whereas for P-MOKE, we used normal incident illumination through a hole in the core of an electromagnet.

## 3. Results

### 3.1. In Situ Characterization

The samples were characterized in situ by LEED and STM. Figure 1 compares LEED patterns for the MgO(111) substrates (a) and magnetite: deposited with no field and in-field, (b) and (c), respectively. The MgO substrate is characteristic of the three-fold symmetry of a single-domain FCC(111) surface. The broad spots reflect the insulating character of the surface and possible charging on the one hand, and moderate surface quality on the other hand, which is the result of low annealing temperatures preventing the faceting of this polar surface [55] and the references therein. The LEED patterns exhibit similar features for both magnetite samples.

As opposed to LEED, the STM images in Figure 2 show meaningful differences in the film morphology that must have been caused by the presence of a magnetic field, whereas all other deposition parameters were kept identical. The surface of the film deposited without a magnetic field has a disordered character, with small (approximately 5–10 nm) grains and an RMS-roughness Rq = 0.40 nm. For the films deposited in the magnetic fields, the grains are much better developed and much bigger (linear dimensions 20–50 nm, on average); however, an RMS-roughness over the entire 400 × 400 nm^2^ image was increased to 0.47 nm. On the other hand, whereas for the no-field sample, Rq only weakly depended on the length scale, for the in-field sample, Rq was reduced to 0.17 nm on the single-grain area. These different morphologies are also seen in the height profiles below the STM images. However, we do not see correlations between the magnetic field direction and STM features, which are rather isotropic with respect to the grain shape and distribution. It should be noted that the bright spots in Figure 2a contribute, to some extent, to the measured roughness. Since we also have micrometer scans (compare Appendix A), we know that these bright spots are characteristic of the no-field sample and they are much less numerous for the in-field sample, although they have a comparable height of up to 3 nm. Unfortunately, we do not find a straightforward explanation for the bright features; they may come from initial impurities on the substrates.

The in situ CEMS measurements were performed for selected samples for verification of the volume film properties; in particular, the stoichiometry. An in situ CEMS spectrum for an uncoated 10 nm magnetite film is shown in Figure 3a. The main spectral features correspond to those of bulk magnetite, as discussed in detail in our previous papers [42,56]. The intense spectral components come from tetrahedral (A-sites) and octahedral (B-sites) iron ions and the red and green sub-spectra, respectively. For stoichiometric magnetite, the A-sites are occupied by Fe^3+^ ions, and twice as many B-sites are occupied by both Fe^3+^ and Fe^2+^ ions. At RT, electron-hopping occurs in the B sublattice on a time scale of approximately 10^−12^ s. Since the characteristic time of the Mössbauer spectroscopy is on the order of 10^−8^ s, the octahedral Fe ions are indistinguishable, and the octahedral sextet B reflects an average charge state of 2.5+. A low-intensity (up to 8%) gray component is associated with irregular surface/interface Fe sites.

An intensity ratio of the spectral components related to the 2:1 occupation of the octahedral and tetrahedral sites reflects stoichiometry. For perfect stoichiometry, the ratio β of the B to A spectral intensity deviates from 2:1 site occupation due to slightly different recoil-free fractions for the tetrahedral and octahedral sites, and several authors have reported β ≈ 1.9 for stoichiometric magnetite [57,58]. On the other hand, a recent thorough analysis reported β = 2 for single crystalline magnetite [59], ideally reflecting the site occupation, and this standard is used in the present study. Consequently, the in situ CEMS spectrum in Figure 3a documents perfect stoichiometry, with β = 2.05, where the fit uncertainty is ±0.1.

Unfortunately, we found that uncoated films are subjected to oxidation towards maghemite after being exposed to the atmosphere for magnetic measurements. Figure 3b,c show spectra for the sample characterized previously in situ (Figure 3a), which were measured ex situ one and six months after preparation, respectively. The oxidation was manifested by a progressing reduction of the sub-spectra intensity ratio. As discussed by Voogt et al. [37], oxidation leads to iron vacancies in the octahedral sublattice, and changes the β ratio. β becomes a sensitive measure of the vacancy parameter δ in the chemical formula of a nonstoichiometric magnetite, Fe_(3−δ)_O_4._ Deriving β from the CEMS spectra, δ can be calculated as $\delta =\frac{2-\beta }{6+5\beta }$. Hence, δ (β) changes from 0 (2) for stoichiometric magnetite to 1/3 (0) for the Fe_2_O_3_ stoichiometry of maghemite. For the spectra taken one and six months after atmosphere exposure, the results change to δ = 0.04 ± 0.005 and δ = 0.12 ± 0.005, respectively. These data can be recalculated to the equivalent film thickness, fully transformed to maghemite—1.6 nm and 4.2 nm, respectively—of a total 10 nm thickness. Because such pronounced oxidation may not leave magnetic properties unaltered, we will further discuss the magnetite films protected before exposure to the atmosphere with a 3 nm MgO coating layer.

### 3.2. Magnetic Properties

CEMS gives information on the local magnetic properties probed by the hyperfine magnetic field B_hf_, where “local” means unaffected by magnetization distribution due to the domain structure or APBs. Neglecting strictly surface effects [60], B_hf_ is a good measure of the magnetic moments. The bulk values of B_hf_ are 49.1 T and 46.0 T at RT for the tetrahedral and octahedral sites, respectively. For the series of the measured stoichiometric 10 nm films, the B_hf_ values were 47.5 ± 0.3 T and 44.0 ± 0.5 T, respectively. This corresponds to only a 4% reduction in the local magnetization and, hence, of the site-specific magnetic moments. This observation indicates that the strong reduction in the magnetic moments derived from magnetization measurements is probably related to the lack of saturation under the applied magnetic field.

The CEMS measurements also provide reliable information on the average direction of spontaneous magnetization. The distribution of the easy magnetization axes and typical geometry of a CEMS experiment for (111)-oriented magnetite films is shown in Figure 4a. One of the easy axes is perpendicular to the surface, whereas the other is at an angle of 70.5° from the surface normal. The angle *θ* between the γ-rays and magnetization is reflected in the line intensities ratio of a Mössbauer sextet, which is 3:*R*:1:1:*R*:3, with $R=\frac{4sin^{2}\theta }{2-sin^{2}\theta }$. In particular, with the γ-rays along the film normal, which was the CEMS geometry of the discussed ex situ measurements, for the two limiting cases of the in-plane and perpendicular magnetization, *R* = 4 and *R* = 0, respectively. For the situation depicted in Figure 4a, magnetization evenly distributed along all easy directions would average *R* to 2.4. It should be noted that the *R* ratio depends on squared sinus, and, hence, the preference for particular easy axes can also be deduced when the macroscopic magnetization averages to zero due to a specific domain structure with up and down or left and right pointing magnetization.

In Figure 4b,c, we show ex situ CEMS spectra for the 10 nm Fe_3_O_4_(111) films on MgO(111) deposited with no field and in-field, respectively. The in-plane magnetic field of 100 mT was applied along the [$1\bar{1}0]$ direction. The hyperfine parameters, isomer shift, hyperfine magnetic field, and intensity ratio of the spectral A and B components are identical for both samples, within experimental error, unambiguously identifying stoichiometric magnetite (compare Appendix A of the Appendix A). However, the films differ in magnetic properties. First, similarly to previous studies, both for (001)- and (111)-oriented magnetite films, for the discussed samples, the magnetization does not lay in the sample plane, as expected when shape anisotropy would be dominating. For the no-field film, the R-value of 2.6 roughly corresponds to the situation wherein all easy magnetization directions are evenly occupied; however, one cannot exclude a canted magnetization at an average angle θ = (62 ± 1)° relative to the film normal. Importantly, a clear influence of the in-plane magnetic field during deposition on the spontaneous magnetization is observed by a decrease of the R-ratio to 1.8 that corresponds to θ = 52°, which signifies an enhancement of the out of plane anisotropy.

An essential complement to the magnetic characterization was MOKE measurements performed in longitudinal (L-MOKE) and polar (P-MOKE) geometry. In the MOKE measurements, a beam of linearly polarized light hits a sample placed in a magnetic field. Reflection from a magnetized surface changes the polarization of the light beam from linear to elliptical, rotates the plane of polarization, and changes the intensity of the reflected light. Changes in the light intensity are recorded by the detector, preceded by an appropriate set of optical elements (analyzer and lenses). Kerr rotation and Kerr ellipticity are approximately proportional to the magnetization of the investigated system.

A selection of the MOKE hysteresis loops is shown in Figure 5. The green L-MOKE loops in Figure 5a,b for the no-field sample, which were measured with the magnetic field along azimuths different by 60°, prove an in-plane anisotropy of the magnetization. Additionally, by measuring the full 360° angular dependence, we found a three-fold (and not the expected six-fold) symmetry of the measured Kerr loops, which signifies in-plane and out-of-plane states of magnetization [61]. Apparently, the in-plane magnetic field did not induce a uniaxial in-plane anisotropy and the anisotropy remains tri-axial. The L-MOKE easy loop for the no-field sample saturates at a moderate magnetic field of approximately 100 mT, with almost 90% saturation in the remanent state. The remanence magnetization value perfectly agrees with the average magnetization angle θ = (62 ± 1)° from the CEMS results, which corresponds to 88% of the saturated magnetization state, indicating a preference for a single-domain state. The hard-direction L-MOKE loop that hardly saturates at 600 mT suggests a strong in-plane anisotropy. However, separating longitudinal and polar contributions to the Kerr signals is necessary to interpret this loop [56]. A P-MOKE loop in Figure 5c for the no-field sample is almost hysteresis-less and has a distinctly hard character. These results provide evidence of a dominating in-plane anisotropy for the no-field sample.

The key observation of the present study is the strong influence of the in-field deposition on the MOKE magnetization loops. The in-plane field applied during deposition changed the loop character enhancing the out-of-plane anisotropy. Additionally, the average magnetization in the remanent state, as judged from the CEMS *R*-factors and remanence in the L-MOKE and P-MOKE loops, was distinctly different, which means a developed domain structure in the remanent state. In parallel, the L-MOKE loops were visibly harder (but narrower), and the P-MOKE curve became hysteretic, with a meaningful 15% remanence. This value can be compared with the value derived based on the corresponding CEMS spectrum: the *R* ratio, in this case, yields approximately 60% of spin components pointing along the perpendicular up and down directions, suggesting a hard perpendicular domain structure.

## 4. Discussion and Conclusions

The observed effect of the external magnetic field applied during MBE growth of ultrathin (111)-oriented magnetite films, albeit very inspiring and promising, is not unexpected, and its anticipation was the base of the present study. The selected material, magnetite, is characterized by strong magnetostriction, and the magnetostrictive effects are known to be enhanced in epitaxial heterostructures [62]. It has been demonstrated that magnetic anisotropy in epitaxially-grown magnetite thin films could be manipulated in situ by applying tunable stress [63]. We have shown that such manipulation is possible in the phase of film growth using MF-assisted MBE. Our STM observations show that the strain modification induced by the external magnetic fields leads to changes in the growth morphology, which seems to be the primary reason for the observed modification of the magnetic anisotropy.

For quantitative determination of the involved anisotropies, we performed simulations of the magnetization curves that best reproduce the measured hard loops. The simulations were based on a simple one-domain model, in which the magnetite layer was parametrized using saturation magnetization *Ms*, effective magnetocrystalline anisotropy constants *K*_1_ and *K*_2_, effective uniaxial anisotropy constants *K_u_*_1_ and *K_u_*_2_, and the shape anisotropy. The simulations showed (for details, see Appendix A) that the in-field deposition contributed to an additional perpendicular uniaxial anisotropy of approximately 5.1 × 10^5^ erg/cm^3^; thus, the order of magnitude of the total uniaxial anisotropy compares to the shape anisotropy and strongly overwhelms the magnetocrystalline anisotropy.

In conclusion, we successfully implemented the magnetic-field-assisted epitaxial growth of magnetite films under UHV. We showed the essential impact of the magnetic field applied during deposition on the magnetic properties of epitaxial layers. Remarkably, a moderate magnetic field of 0.1 T induced a sizable effect in the magnetic anisotropy. This suggests that the effect of the external magnetic field during deposition on the magnetic properties is magnetostrictive because other mechanisms, such as a direct influence of the magnetic field on the growth kinetics, would require much higher fields [6]. We are far from fully understanding the role of the magnetic field during epitaxial growth, and such aspects as different in-plane field orientations, as well the effect of perpendicular magnetic fields, should be further studied. This notwithstanding, the present results pave the way for tailoring magnetic (and maybe structural) properties of more complex epitaxial heterostructures, such as exchange bias systems or spin valves.

## Figures and Tables

**Figure 1 materials-16-01485-f001:**
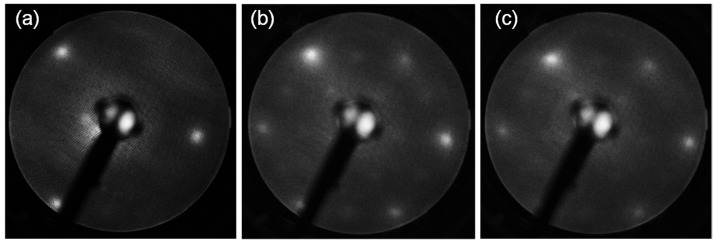
LEED patterns for the MgO(111) substrates (**a**) and magnetite: deposited with no field (**b**) and in-field (**c**). The electron energy is 100 eV.

**Figure 2 materials-16-01485-f002:**
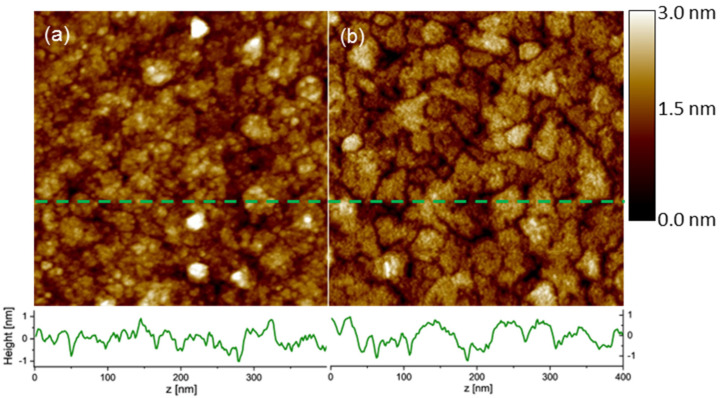
STM images of the 10 nm Fe_3_O_4_(111) films on MgO(111) deposited with no field (**a**) and in-field (**b**). The scan size is 400 × 400 nm^2^. The height scale is the same for both images. The plots below the STM images show the height profiles along the marked dashed lines.

**Figure 3 materials-16-01485-f003:**
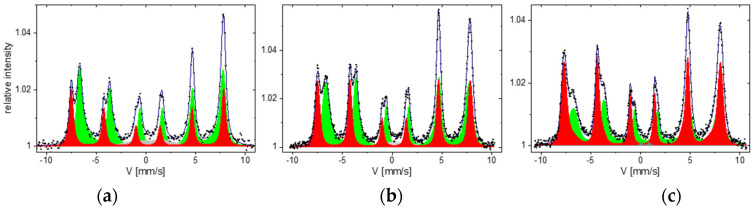
CEMS spectra of a 100 Å magnetite film measured in situ (**a**) and ex situ, after one and six months after exposure to the atmosphere (**b**,**c**), respectively.

**Figure 4 materials-16-01485-f004:**
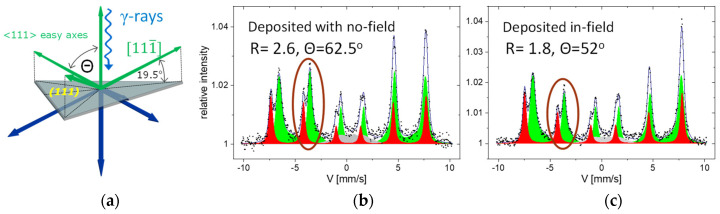
(**a**) Configuration of the easy magnetization axes for the (111)-oriented magnetite films. γ-ray illumination direction is indicated in the ex situ CEMS experiments. (**b**,**c**) CEMS spectra for the 10 nm Fe_3_O_4_(111) films on MgO(111) deposited with no-field and in-field, respectively.

**Figure 5 materials-16-01485-f005:**
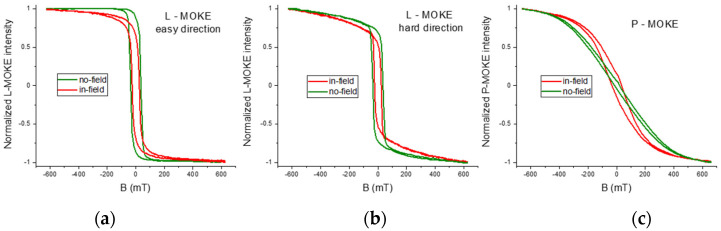
MOKE hysteresis loops measured for the 10 nm Fe_3_O_4_(111) films deposited with an in-plane magnetic field applied along the $\left(1\bar{1}0\right)$ direction and with no field (red and green curves, respectively). (**a**,**b**) L-MOKE loops with the magnetic field applied along in-plane easy and hard directions, respectively; (**c**) P-MOKE loops.

## Data Availability

The data presented in this study are available on reasonable request from the corresponding author.

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
