# Peer review of "Magnetic-Field-Assisted Molecular Beam Epitaxy: Engineering of Fe3O4 Ultrathin Films on MgO(111)"

_materials, 2023, doi:10.3390/ma16041485_

Round 1

Reviewer 1 Report

Comments- This manuscript studies Fe3O4 ultrathin film deposited on a MgO

substrate. This study seemed exciting and might be suitable to publish after minor

revision. My comments are given below.

1. How the in-plane and out-of-plane magnetizations are related to the type of substrate and their

orientation also, as mentioned in lines 151-152.

2. Is the spectral intensity ratio obtained from the CEMS, i.e. 2:1, related to the occupancy

of Fe at A and B sites, respectively?

Reviewer 2 Report

In the present manuscript, the author presents a detailed study of magnetic-field-assisted epitaxial growth of magnetite films under UHV. Particularly, it is shown that a moderate magnetic field can induce a sizable effect in the magnetic anisotropy during evaporation.

Overall, I find the experimental results interesting and the topic is undoubtfully new. However, I still have some questions about the deposition process of the sample. If my concerns can be addressed satisfactorily, I would recommend its publication in the revised manuscript.
Below, I describe all this in detail as well as specify the points to address to improve the consistency/clarity of the manuscript.

1. First paragraph needs to add proper references.

2. How close are the two samples in the dual sample stage? How can the author calculate or estimate the effective magnetic field that is applied to the target sample? The author should elaborate more on this point since it is the fundamental part of the whole manuscript.

3. What is the orientation of the in-plane magnetic field? Would the author be able to tune the in-plane field direction or How do the authors compare it with this direction with the feature for example the one shown in the STM topography in Figure 2?

4. All signal in Figure 1 need to be high lightened.

5. Have the authors characterize the roughness of the bright spot in Figure 2a? Is it comparable to the roughness in Figure 2b? Having micrometer size STM topography images and comparing is sometimes hard to convince people. The author should elaborate more evidence or lineprofiles with similar length to improve the clarity of this point.

6. I don't think it would be appropriate to compare the roughness of single grain area with multi grains in without field sample. What would the roughness for figure 2b be if all grains are considered?

7. a/b/c are not labelled properly in Figure 3.

8. Most importantly, does the author think the magnetic properties depends on the in-plane field direction or not? The author should elaborate more discussion in the conclusion part.
